# U-47700 and Its Analogs: Non-Fentanyl Synthetic Opioids Impacting the Recreational Drug Market

**DOI:** 10.3390/brainsci10110895

**Published:** 2020-11-23

**Authors:** Michael H. Baumann, Graziella Tocco, Donna M. Papsun, Amanda L. Mohr, Melissa F. Fogarty, Alex J. Krotulski

**Affiliations:** 1Designer Drug Research Unit, Intramural Research Program, National Institute on Drug Abuse, National Institutes of Health, 333 Cassell Drive, Suite 4400, Baltimore, MD 21224, USA; 2Department of Life and Environmental Sciences, University of Cagliari, Cittadella Universitaria di Monserrato, Monserrato, 09042 Cagliari, Italy; toccog@unica.it; 3Toxicology Department, NMS Labs, 200 Welsh Road, Horsham, PA 19044, USA; Donna.Papsun@NMSLABS.COM; 4Center for Forensic Science Research and Education, Fredric Rieders Family Foundation, 2300 Stratford Ave, Willow Grove, 19090 PA, USA; Mandi.Mohr@cfsre.org (A.L.M.); Melissa.Fogarty@cfsre.org (M.F.F.); Alex.Krotulski@cfsre.org (A.J.K.)

**Keywords:** analytical chemistry, forensic toxicology, mu-opioid receptor, opioid pharmacology, U-compounds

## Abstract

The recreational use of opioid drugs is a global threat to public health and safety. In particular, an epidemic of opioid overdose fatalities is being driven by illicitly manufactured fentanyl, while novel synthetic opioids (NSOs) are appearing on recreational drug markets as standalone products, adulterants in heroin, or ingredients in counterfeit drug preparations. *Trans*-3,4-dichloro-*N*-[2-(dimethylamino)cyclohexyl]-*N*-methylbenzamide (U-47700) is a prime example of a non-fentanyl NSO that is associated with numerous intoxications and fatalities. Here, we review the medicinal chemistry, preclinical pharmacology, clandestine availability, methods for detection, and forensic toxicology of U-47700 and its analogs. An up-to-date summary of the human cases involving U-47700 intoxication and death are described. The evidence demonstrates that U-47700 is a potent μ-opioid receptor agonist, which poses a serious risk for overdosing and death. However, most analogs of U-47700 appear to be less potent and have been detected infrequently in forensic specimens. U-47700 represents a classic example of how chemical entities from the medicinal chemistry or patent literature can be diverted for use in recreational drug markets. Lessons learned from the experiences with U-47700 can inform scientists, clinicians, and policymakers who are involved with responding to the spread and impact of NSOs.

## 1. Introduction

The misuse of opioid drugs is a serious public health concern [1]. In particular, a dramatic rise in opioid-induced intoxications and fatalities is being driven by illicitly manufactured fentanyl appearing in non-medical (i.e., recreational) drug markets worldwide [2,3]. An additional disturbing trend is the emergence of novel synthetic opioids (NSOs) that are encountered as standalone products, adulterants in illicit heroin, or constituents of counterfeit pain medications [4,5,6]. Here, we define NSOs to include fentanyl analogs and non-fentanyl-related compounds that function as potent and efficacious μ-opioid receptor (MOR) agonists. NSOs present complex challenges for public health and safety since these drugs are often more potent than heroin or morphine, and they are not detected by standard toxicology drug testing methods [7,8]. Most NSOs are not “new” chemical entities, rather, they are resurrected from older patent and medicinal chemistry literature where their potency and efficacy are often reported. Since the 1970s, pharmaceutical companies and medical researchers have sought to discover and develop new analgesic agents with reduced propensity for undesirable side-effects, such as tolerance, dependence, and respiratory depression. Some of these potential medications, belonging to different chemical classes, have been diverted as NSOs to the recreational drug supply [9,10]. In this regard, phenampromide [11], MT-45 [12,13], AH-7921 [14,15], U-47700 [16,17], and U-50488 [18,19] are notable examples of structurally distinct analgesic compounds that never advanced to human clinical trials (see Figure 1 for chemical structures). U-47700 and U-50488 are part of an *N*-substituted benzamide and 2-phenylacetamide series of compounds that was explored by the Upjohn Company in the 1970s and 1980s as potential therapeutic agents. Drugs in these series can be referred to as “U-compounds” (or “U-series” compounds, or informally as “Utopioids”), with the “U” attributable to their company of origin. 

It is hypothesized that U-compounds emerged as NSOs due to their desirable opioid-mediated effects (i.e., euphoria and pain relief), together with ease of synthesis and availability of precursor chemicals or starting materials [20,21]. Clandestine drug products containing U-47700 are nicknamed “pink”, “U4”, or “pink heroin” by drug distributors and consumers, and more recently this drug was identified as a constituent in the potent opioid cocktail “gray death” [22]. Studies using laboratory animals show that U-47700 is about 10-times more potent than morphine as an analgesic agent [23], and this enhanced potency could be responsible for the intoxications and mortality associated with the compound [24,25,26,27]. Published reviews about U-47700 have covered its history and diversion [20], synthetic schemes and structure activity relationships (SAR) [21], as well as clinical symptoms [28]. Here, we provide an updated review of the medicinal chemistry, preclinical pharmacology, clandestine availability, methods for detection, and forensic toxicology of U-47700 and selected analogs. A complete summary of the human casework involving U-47700 intoxication and death is included.

## 2. Medicinal Chemistry

All of the NSOs depicted in Figure 1 are non-fentanyl-related compounds that share a strategic 1,2-ethylene diamine core scaffold with vast potential for side chain substitutions and rearrangements, thereby serving as a template for a variety of structurally related analogs [29]. With the exception of MT-45, all of these compounds possess two nitrogen atoms with very different chemical and electrochemical characteristics, so they can be better defined as *N*-(2-ethylamino) amides. More specifically, these compounds possess one sp_3_ hybridized trisubstituted nitrogen that is strongly basic, and another sp_2_ hybridized amide nitrogen with no electron pair available for protonation. Interestingly, embedding the *N*-(2-ethylamino) amide into a rigid cyclohexyl scaffold, as with the U-compounds, results in a dramatic improvement in opioid receptor affinity and selectivity when compared to AH-7921 [17,30]. U-47700 was one of the first U-compounds identified during medication development efforts aimed at finding novel non-addictive opioid analgesics, and it showed high affinity for MOR in receptor binding assays. Further substantial modification to the core structure of U-47700 gave rise to U-50488, which displays high selectivity for the κ-opioid receptor (KOR) over MOR. U-50488 and its analogs are prototypical KOR agonists. Although U-50488 and related compounds were never developed for human use due to undesirable side-effects (e.g., sedation, dysphoria, and diuresis), these compounds remain important research tools for studying the role of KOR in animal models [30,31]. A number of SAR studies involving U-compounds have been carried out to better understand the influence of various structural modifications on biological activity [32,33,34]. The generic U-compound scaffold and its potential for structural modification(s) is depicted in Figure 2.

Szmuszkovicz first observed that the introduction of a methylene spacer between the aromatic ring and carbonyl moiety of U-47700 is a key determinant for the MOR to KOR switch in selectivity, terming this the “eastern methylene group effect” [30]. Based on the absence or presence of the methylene group, U-compounds can be broadly categorized into two major groups: (1) the U-47700 series MOR compounds and (2) the U-50488 series KOR compounds. Figure 3 and Figure 4 show representative chemical structures for both series. Data reported in Table 1 and Table 2 demonstrate how this simple structural modification makes a notable difference in terms MOR/KOR binding selectivity [35]. From a 3-dimensional structure perspective, the absence of the methylene spacer in U-47700 allows the basic nitrogen and aromatic ring to assume a spatial orientation similar to that of morphine, which confers selective interaction with MOR [16]. Conversely, the addition of the methylene spacer in U-50488 induces a “bending” of the benzene ring, which causes variation in the spatial relationship between the aminoamide and the aryl moiety. Moreover, together with the cyclohexyl scaffold, the methylene enables the portion of the molecule between the basic tertiary nitrogen and the amide to adopt the torsional angle of 60° in a low-energy conformation necessary for a selective interaction with KOR [36].

As previously noted, the inclusion of the 1,2-ethylendiamine moiety into a cyclohexane ring represents a key structural alteration for U-compounds when compared to AH-9721. In particular, cyclization affords rigidity to the scaffold and generates stereocenters at C1 and C2 that yield four stereoisomers: *cis* (1R,2S), *cis* (1S,2R), *trans* (1R,2R), and *trans* (1S, 2S). Importantly, for both the U-47700 and U-50488 series compounds, only the *trans* stereoisomers display opioid activity [30]. However, the absolute configuration of the biologically active *trans* isomer differs between the U-47700 series and U-50488 series. For the U-47700 series, the *trans* (1R,2R) stereoisomer has the greatest affinity and selectivity for MOR, whereas in the U-50488 series, the *trans* (1S,2S) stereoisomer of U-50488 (also referred to as U-50488H) has the greatest affinity and selectivity for KOR [30,33,36,37,38]. Molecular mechanics and *ab initio* calculations carried out with the U-47700 enantiomers found that the *trans* (1R,2R) stereoisomer fits perfectly into the MOR binding site, even with a slightly reduced strain energy [16]. A recent study by Hsu et al. [39] examined the in vitro MOR activity of the U-47700 stereoisomers using a cyclic adenosine monophosphate (cAMP) accumulation assay and confirmed that the *trans* (1R,2R) stereoisomer (EC_50_ = 8.8 nM) is much more potent than the *trans* (1S,2S) stereoisomer (EC_50_ ~1 μM). Investigations into the stereochemistry of U-50488 demonstrate that both the *cis* isomers display a complete loss of affinity for KOR but show a corresponding increase in affinity for other receptor systems, including sigma receptors [40] and D2-dopamine receptors [38]. Collectively, the findings reveal the critical role of stereochemistry in the opioid receptor activity for compounds of the U-4700 series and U-45088 series.

Substitution on the cyclohexane ring may markedly affect both receptor affinity and selectivity for U-compounds. Molecular modeling studies suggest that the cyclohexane ring is implicated in key interactions with the hydrophobic portion of the opioid receptor binding pocket. Within the U-47700 series, replacing the chlorine substituents on the phenyl ring greatly influences the affinity for MOR. Removal of the chlorine atom from the *meta*-position generates compound U-48520, which has a MOR binding affinity (K_D_ = 200 nM) much lower than that of U-47700 (K_D_ = 5.3 nM). The replacement of both chlorine atoms with less electronegative bromine atoms, as with compound U-77891, results in a slight enhancement of the MOR affinity (K_D_ = 2 nM). This trend suggests that the benzene ring is most likely involved in electronic function rather than hydrophobic interactions. The introduction of an oxaspiro ring *para* to the amide nitrogen converts U-50488H to U-62066 and U-69593, analogs which are highly selective for KOR over MOR. More precisely, U-69593 is more KOR selective while its chlorinated analogue U-62066 is less selective at the KOR but more potent as an analgesic [18,41]. In addition to increasing the MOR affinity, it should be noted that the halogens added to the 3- and 4- positions serve to protect the U-compounds from metabolic hydroxylation, which could enhance the duration of their in vivo effects [30].

The spatial relationship between the aryl moieties and the aminoamide is a critical determinant for the MOR versus KOR binding affinities for U-compounds. In this regard, both the amino and amido *N*-substituents are decisive for contact with opioid receptor binding sites [30,31]. In particular, the sterically unhindered *N,N-*dimethylamino group greatly enhances the MOR affinity, as in U-47700. By contrast, embedding that same basic nitrogen into a cyclic pyrrolidine ring greatly enhances the KOR affinity, as in U-50488. MOR/KOR selectivity is influenced by the *N*-amido substituents as well. For example, the removal of the methyl group to a secondary amide, as in U-47109, results in a 50-fold loss in MOR affinity when compared to U-47700 (although see Hsu et al. [39]). In conclusion, compounds from both the U-47700 series and the U-50488 series possess two hydrophobic sites (cyclohexyl and phenyl moieties) and two polar sites (amino and amido nitrogens). The mutually exclusive spatial orientation (i.e., methylene spacer) of the U-47700 series versus the U-45088 series is what predominantly tips the scale in favor of MOR versus KOR binding epitopes, respectively.

## 3. Preclinical Pharmacology

None of the U-compounds discussed above were ever approved for clinical use as a medication, and only a few studies evaluated the pharmacological effects of these agents after controlled administration in humans. The U-50488 series compound, spiradoline (U-62066), was tested in humans and produced dose-related analgesic and diuretic effects, but significant side effects, such as sedation, dysphoria, and hallucinations, precluding its clinical development [42]. On the other hand, the in vivo pharmacology of U-50488 series compounds has been well studied in animal models, and these agents have been used as prototypical KOR agonists in preclinical medication development efforts [30]. Far fewer studies have examined the in vivo pharmacology of the U-47700 series compounds. Cheney et al. [16] reported one of the first SAR characterizations of U-47700 and its analogs by examining the inhibition of [^3^H]naloxone binding in rat brain homogenates and analgesic activity in the mouse radiant tail flick test. These investigators found that U-47700 displays an agonist-binding affinity at the rat MOR, which is about 8-fold weaker than that of morphine. Interestingly, U-47700 had an ED_50_ of 0.2 mg/kg in the mouse tail flick test whereas morphine had an ED_50_ of 1.5 mg/kg. Thus, when compared to morphine, U-47700 exhibits a lower affinity for MOR but a higher potency as an analgesic agent. The work of Cheney et al. was crucial for identifying the relationship between the in vitro binding affinity at MOR and in vivo analgesic potency of U-47700.

Following its initial characterization in the 1980s, the investigation of U-47700 remained dormant for more than three decades, until the compound appeared on recreational drug markets as an NSO, beginning around 2014. This occurrence spurred renewed interest in the drug from many different perspectives. It should be mentioned that the original studies examining the effects of U-compounds, and related opioid agents, were carried prior to the cloning of opioid receptor subtypes, and the subsequent development of genetically engineered mice lacking opioid receptor gene splice variants. With recent advances in molecular biology, the mechanisms of opioid drugs can be investigated with greater clarity and detail [43]. To this end, Baumann et al. [44] recently evaluated the in vitro pharmacological effects of U-47700 in Chinese Hamster ovary (CHO) cells transfected with murine MOR, KOR, or δ opioid receptors (DOR). Specifically, these investigators examined opioid receptor affinity using inhibition of [^125^I]IBNtxA binding and examined opioid receptor function using the [^35^S]GTP-γ-S assay. The binding results showed that U-47700 is MOR selective, with Ki values of 57 nM (MOR), 653 nM (KOR), and 1105 nM (DOR). By comparison, morphine had a Ki value of 5 nM at MOR, indicating that U-47700 exhibits a 10-fold lower affinity than morphine at the mouse MOR. Results from the [^35^S]GTP-γ-S assays mirrored the binding results and revealed that U-47700 functions as a full agonist at all opioid receptor subtypes, with high selectivity for MOR over KOR and DOR. Baumann et al. [44] also compared the in vivo analgesic effects of U-47700 to those of morphine and its fentanyl analog, butyrylfentanyl. The data depicted in Figure 5 show that U-47700 displays an ED_50_ of 0.21 mg/kg in the mouse tail flick test whereas morphine displays an ED_50_ of 2.5 mg/kg. The analgesic effect of U-47700 was reversed by the opioid antagonist naloxone and was absent in genetically engineered mice lacking a functional MOR, reinforcing the critical involvement of the MOR in mediating drug effects. Taken together, the results of Baumann et al. [44] confirm and extend the original findings of Cheney et al. [16] that U-47700 is a potent MOR-selective agonist. 

It is noteworthy that U-47700 displays a lower binding affinity than morphine in both rat and mouse MOR isoforms, yet U-47700 has a much higher analgesic potency than morphine in both species. The precise reason(s) why U-47700 is more potent in vivo than predicted by its in vitro binding affinity for the MOR is not known, but could be related to enhanced brain penetration of the drug owing to its higher lipophilicity. Based on physiochemical properties alone, U-47700 (clogP = 4.09) is much more lipophilic when compared to morphine (clogP = 0.57). Other possibilities include differences in intrinsic activity or cell-signal amplification mechanisms between U-47700 and morphine. The findings comparing U-47700 to morphine demonstrate that in vitro results alone are insufficient to characterize the pharmacological effects of opioid compounds, and stress the critical importance of determining in vivo drug potency when considering the risk assessment of NSOs as they appear on recreational drug markets.

There is little information available about the biotransformation or in vivo clearance of the U-47700 series compounds. Based on the chemistry of the compounds, putative pathways for metabolism include *N*-dealkylation and hydroxylation of the cyclohexyl ring. Krotulski et al. [45] first identified *N*-desmethyl-U-47700 and *N,N*-didesmethyl-U-47700 as the major metabolites of U-47700 in urine samples obtained from human casework. In the same study, *N*-desethyl-U-49900 was identified as the primary metabolite of U-49900, whereas *N,N*-didesethyl-*N*-desmethyl-U-49900 was the most abundant species in urine. There is the potential for shared metabolites between U-47700 and U-49900, as in the case of 3,4-dichloro-*N*-(2-aminocyclohexyl)-*N*-methylbenzamide, due to structural similarities within this group of substances [45]. The presence of *N*-desmethyl-U-47700 and *N,N*-didesmethyl-U-47700 have been confirmed in blood samples obtained from human subjects exposed to U-47700, confirming that these metabolites are the major species formed [27,46]. *In vitro* and in vivo studies were performed to identify the metabolites of U-48800, and 14 metabolites were tentatively identified with *N*-dealkylation, hydroxylation, and combinations of the two mechanisms, suggesting these as the primary metabolic pathways [47].

There is a single case report of U-47700 pharmacokinetics in humans that involved the withdrawal of repeated blood samples from a hospitalized patient who overdosed following the ingestion of U-47700 and eventually died [48]. It was found that U-47700 displays an elimination half-life of approximately 6.5 h, but there are caveats to these clinical results, including a lack of C_max_ determination and co-ingestion of the benzodiazepine, flubromazepam, which could affect opioid drug clearance. Due to the absence of controlled clinical studies with U-47700 and other NSOs, animal models can be used to fill an important void in the available data. To this end, Truver et al. examined the pharmacokinetic and pharmacodynamic effects of U-47700 administered to male rats bearing indwelling jugular catheters for repeated blood sampling [49]. The data of Truver et al., depicted in Figure 6, show that U-47700 displays dose-proportional kinetics in rats, with an elimination half-life of approximately 1–2 h, much shorter than that observed in the human case. The shorter half-life of U-47700 in rats probably reflects the generalized phenomenon of faster drug clearance in rodents when compared to humans [50]. Importantly, the plasma C_max_ values for U-47700 reported in rats (i.e., 50–150 ng/mL) overlap with the blood concentrations measured in non-fatal U-47700 intoxications in humans [51,52], suggesting the rat model has translational value. Truver et al. also reported that *N*-desmethyl-U-47700 and *N,N-*didesmethly-U-47700 are formed in rats, but these metabolites exhibit a much slower clearance when compared to the parent compound. U-47700 produces analgesia (ED_50_ = 0.5 mg/kg, s.c.) and catalepsy (ED_50_ = 1.7 mg/kg) in rats, and these pharmacodynamic effects are positively correlated with plasma concentrations of U-47700 and *N*-desmethyl-U-47700. However, radioligand binding studies using inhibition of [^3^H]DAMGO binding in rat brain tissue demonstrate that the affinity of *N*-desmethyl-U-47700 (Ki = 206 nM) and *N,N*-didesmethyl-U-47700 (Ki = 4080 nM) at the MOR are much less than that of U-47700 (Ki = 11 nM). Collectively, the findings from rats suggest that U-47700 is metabolized by *N-*dealkylation, similar to the situation in humans, but the dealkylated metabolites probably do not contribute significantly to the pharmacodynamic effects of the drug. 

## 4. Clandestine Availability

The emergence of U-47700 on recreational drug markets dates as far back as 2014, with published toxicology case reports following in 2016. During the last 7 years, U-47000 has impacted national media headlines as one of the most dangerous NSOs, responsible for many fatalities. As a result, the drug is now declared illegal in many countries worldwide [53,54]. Several U-compounds have been sold recreationally, via online crypto drug markets and on the street, typically in powder or pill form, and less commonly in nasal sprays and e-liquids. U-47700 has also been reported as the primary substance in counterfeit pharmaceuticals, a common occurrence with NSOs. In March 2016, officials seized 500 pills in Lorain County, Ohio, United States (US), which visually appeared to be oxycodone pills; chemical analysis confirmed the pills contained U-47700 [55]. The first quarter of 2016 was the first time U-47700 was chemically identified in the US [56]. In June 2017, there was a cluster of opioid overdoses in Georgia, US, related to counterfeit Percocet^®^ pills; chemical analysis of the pills revealed U-47700 and a novel fentanyl analog, cyclopropylfentanyl [57]. U-47700 was also detected in clandestine tablets seized in Brazil [58]. U-47700 has been reported in case reports and/or seized materials in several countries, including the US, United Kingdom, Sweden, Belgium, Italy, Czech Republic, Finland, and Brazil [54,59]. In the US, all but nine states (AK, HI, ME, MT, NE, NM, SD, VT, and WY) reported at least one U-47700 case in 2017 in data available from the US Drug Enforcement Administration (DEA) [60]. 

Due to the perceived popularity of U-47700 with recreational drug users, the natural progression was for chemically similar compounds to be explored for their viability on the illicit drug market. As mentioned, the U-compounds have never been studied in humans and are not registered for medical use; however, their opioid agonist properties have been reported and proved enticing as alternatives to routinely encountered illicit opioids, such as heroin. An added advantage of evading routine drug screens and circumventing drug laws was also a factor. Nevertheless, in addition to the desired effects of analgesia and euphoria, the dangerous adverse effects of opioids are to be expected, including sedation, depressed consciousness, and respiratory depression, which can be fatal. U-49900 and U-51754 emerged in 2016, with U-48800, U-47931E (bromadoline), and U-77891 following in 2017. Other U-compounds confiscated include U-50488, 3,4-methylenedioxy-U-47700, and isopropyl-U-47700, between 2017 and 2018 [20,61]. The Center for Forensic Science Research and Education (CFSRE) through its New Psychoactive Substances (NPSs) Discovery program reported the detection of a number of U-compounds in either seized drug materials or toxicology samples (Table 3). New compounds of this class continue to emerge, but none of the compounds encountered thus far have achieved the same level of popularity or threat to public health when compared to U-47700, in some part due to the now overwhelming popularity of fentanyl itself in the recreational market. Fewer countries have reported detections of other compounds in the series, but U-47700 continues to dominate in seized materials and/or toxicology casework, although at significantly lower levels compared to its peak popularity.

## 5. Methods for Detection

The analytical identification of a specific NPS in forensic samples is critical for understanding the overall prevalence and spread of the emerging substance, whether it is detected in seized drug material or toxicology samples. Compound identification is always the first step in interpreting the role of a specific NPS in casework or other circumstances. However, detecting NPSs, and specifically NSOs, in forensic casework is challenging due to the novelty of these drugs and the limited capabilities of the legacy instrumentation. The two largest challenges for detecting U-compounds are (1) availability of standard reference material for comparison and/or confirmation and (2) utility of instrumentation to provide sufficient specificity, sensitivity, and discovery potential.

With forensic testing, two analyses (or screening and confirmation techniques) are required for reporting of forensically defensible results, as outlined by governing bodies and accrediting agencies. Typically, this requires the availability of a highly purified reference material for the comparison of analytical data. Reference materials are chemically verified synthetic products that are generally free from impurities and prepared at a known concentration; important aspects for both qualitative and quantitative confirmations. The availability of reference material for NPSs, and especially their metabolites or isomers, can significantly lag behind the initial appearance of a substance on the market. Forensic laboratories heavily rely on synthetic organic chemists and reference material manufacturers (e.g., Cayman Chemical Company) for these standards, and the entities must work together for timely and accurate results. Once reference materials are available, screening and confirmation methods can be developed or modified to include the new substance.

The testing of seized drug materials has become a standard, streamlined process in forensic chemistry or criminalistics laboratories. The overwhelming majority of seized drug testing for drugs of abuse, including NPSs, is conducted using gas chromatography mass spectrometry (GC-MS), both for screening and confirmation, but using differing chromatographic methods or conditions. GC-MS bodes well for seized drug testing due to the large amount of sample available for testing (milligram to gram quantities), as well as its implementation as a non-targeted technique requiring less input and development. Standard reference materials are used to build robust and transferrable library databases, which include retention time information and mass spectra, the equivalent to a chemical fingerprint. Less commonly, seized drug testing laboratories will use more specific instrumentation for the discovery and confirmation of new substance (e.g., for the U-compounds). This more esoteric testing is generally conducted using high-resolution mass spectrometry (HRMS) to generate a proposed formula for the unknown substance and to elucidate a possible structure based on fragmentation (if applicable); subsequent confirmation of the structure is then performed using nuclear magnetic resonance (NMR) spectroscopy. Unlike GC-MS and HRMS techniques, which use chromatography as a separation step, NMR analysis requires a pure substance and cannot be performed using street-level samples, which can contain multiple constituents.

With forensic toxicology samples, screening by HRMS, typically using time-of-flight (TOF) or quadrupole time-of-flight (QTOF) mass analyzers, offers advantages over more traditional screening techniques (e.g., immunoassay-based technology and targeted mass spectrometry assays). HRMS methods are most successfully designed in non-target acquisition mode, meaning these methods detect all analytes present within the analysis over a certain threshold (e.g., peak intensity) or within given criteria (e.g., mass range). Because the amount of drug in a toxicology sample is generally much smaller than the complex background species of the matrix, HRMS allows for a more accurate distinction between drugs and unwanted molecules. HRMS instruments also offer increased sensitivity over immunoassays and GC-MS instruments.

HRMS is very useful for identifying known and unknown compounds, especially when considering the lack of reference standards and the advances in retrospective data-mining. Forensic laboratories are increasingly moving towards HRMS-based platforms for screening purposes due to these aforementioned advantages [62]. HRMS capabilities have become a valuable tool in forensic laboratories, particularly for the early detection of NPS, either during discovery efforts or general screening [63,64,65]. HRMS records accurate mass data for all compounds present in the forensic sample, essentially creating a sample-specific record that can be archived. The data can then be processed with a library database both at the time of original analysis and retrospectively with an updated library (i.e., data-mining), the latter of which does not require the need for re-extraction or re-analysis of samples. 

Liquid chromatography with tandem mass spectrometry (LC-MS/MS), a targeted technique, is the most common technique used for confirmation testing of U-compounds due to higher sensitivity and adequate specificity in relation to HRMS. However, development of LC-MS/MS confirmation methods is dependent on the availability of reference standards for the targeted acquisition, as mentioned previously [66]. 

Retrospective HRMS data analysis was used to successfully identify designer benzodiazepines after the initial detection of only U-47700 in an accidental overdose case [62]. In addition, the re-analysis of biological sample extracts (a process called sample-mining), using liquid chromatography–quadrupole time-of-flight mass spectrometry (LC-QTOF-MS) coupled with an extensive database containing more than 800 compounds, was successfully implemented for the first-time discovery of two novel U-compounds, isopropyl-U-47700 and 3,4-methylenedioxy-U-47700, among a variety of other NPSs [67]. In addition to detecting new substances (e.g., NSOs and their metabolites) that were not targeted in the original screen, a retrospective analysis of the screening data can be utilized to develop a timeline for the emergence of NSOs. For example, a retrospective data analysis of blood samples screened by LC-QTOF-MS determined that the total lifespan of U-47700 in Finland lasted approximately 2 years, with the first fatality occurring in October 2015 and the last fatality occurring in May 2017 [68]. 

There are several published methods for the confirmation of U-compounds in various biological matrices, with the majority of methods developed for blood, serum/plasma, and/or urine [25,27,69,70,71,72,73]. Two major considerations when developing confirmation methods for U-compounds are sensitivity (or predicted concentration range) and isobaric species (different compounds with the same molecular formula). Isomeric pairs exist for both parent drugs and metabolites (e.g., U-48800 vs. U-51754, U-49900 vs. isopropyl-U-47700 vs. propyl-U-47700, and U-47700 vs. *N*-desethyl-U-49900), and these pairs require special attention and chromatographic separation since their mass data are similar or identical. Methods have also been developed for the detection of U-compounds in alternative matrices, such as oral fluid and hair [74,75,76]. U-47700 was identified in oral fluid samples after suspected heroin use [77], as well as hair in both an intoxication event and a fatality [51,78]. 

There is limited information available for understanding the effects of postmortem redistribution on the reported concentrations for U-compounds. Postmortem redistribution is a phenomenon that occurs after death where drugs migrate in the body, causing higher than expected concentrations in some biological matrices (e.g., central blood). This is a known occurrence with the synthetic opioid fentanyl [79]. There is one case report where both central and peripheral blood were analyzed for U-47700 after a fatality; central blood was reported to be 340 ng/mL, whereas peripheral blood was 190 ng/mL. The authors speculated there could be concern for postmortem redistribution of U-47700, with the understanding that the proposal was based on the results from one case [80]. Subsequently, the postmortem redistribution of U-47700 was studied in 10 fatalities from the New York City Office of the Chief Medical Examiner. Vitreous humor and brain were determined to be suitable alternatives to testing femoral blood samples, with brain tissues reporting the highest concentrations of analyte [81]. U-47700 has been reported in vitreous humor, brain, liver, and urine from case reports as well [82]. 

The chemical stability of a substance in biological fluids also influences the interpretation of the final reported concentration. Truver et al. performed a stability study of U-47700, U-48800, and U-50488 at various temperatures over a 36-week period [83]. Most analytes proved stable at high concentrations. At a concentration of 80 ng/mL, U-47700 and U-49900 were deemed unstable at 252 days, while U-50488 remained stable throughout the 36-week period. U-50488 demonstrated instability at a low concentration (0.75 ng/mL) at room temperature conditions after 28 days and exhibited a 26% loss at 28 days when stored at 35 °C. 

Once the U-compounds are detected and reported, forensic toxicologists are tasked with attempting to interpret the results and the concentrations of the intoxicants, especially in medicolegal investigations. However, particularly in cases of emerging substances, the lack of published information regarding pharmacology and toxicology pose formidable challenges. First and foremost, laboratories need to have the capabilities to perform screening and confirmation testing for emerging substances, such as the U-compounds. Without routine screening and quantitative confirmation testing on the parent drug (and metabolites in a variety of matrices), prevalence for the compounds will continue to be poorly characterized. As most NSOs are rarely studied in humans, any description of adverse effects is necessarily derived from case studies and the assertation of toxicity is derived from autopsy findings paired with medical examiner and toxicologist interpretation. However, poly-drug use, or detection of multiple drugs in biological specimens from a single donor, is frequently encountered. Attempting to ascertain the effects of a single substance on the cumulative effect of drug mixtures is extremely difficult, regardless of what concentration(s) may be measured. In addition, the combined use of more than one NSO is commonly reported, including one case from Quebec that reported 17 different NSOs, including U-47700, U-47931E, *N*-methyl U-47931E, U-49900, and U-48800 [84]. While the body of knowledge for the U-compounds is increasing, there is still much more to be explored, including information about matrix ratios, accurate determination of postmortem distribution, half-lives, more detailed metabolism, more complex scenarios for stability, etc. As stated by Frisoni et al., it is imperative that forensic pathologists and toxicologists work together to properly identify these cases of synthetic opioid intoxication in hopes of combatting the public health concerns posed by NPSs [4].

## 6. Forensic Toxicology

Although there are several U-compounds listed in pharmaceutical patents, the recreational drug market tends to filter prospective narcotic drug candidates by selecting those that have potent MOR binding and activation. Furthermore, drug users have a substantial impact on the popularity and prevalence of a particular substance, as they share their experiences on internet drug forums and provide feedback on internet drug sites and with clandestine manufacturers [85]. Overall, this means that there is no set correlation between NSO introduction onto the drug market and assurance of widespread adoption by recreational users. The lifespan of a particular NSO is further impacted by the legal status of the drug, as the demand for a substance will typically decrease in response to its scheduling to an illegal or controlled status. 

U-47700 was the first U-compound to infiltrate the recreational drug market beginning in 2014. The popularity of U-47700 increased in 2015 and 2016. Between July and December 2016, U-47700 was reported in 0.8% of the 5152 opioid overdose deaths from West Virginia, Ohio, and Wisconsin in a ten-state study participating in the Centers for Disease Control and Prevention’s (CDC’s) Enhanced State Opioid Overdose Surveillance (ESOOS) [86]. While this number seems low, it is significant when comparing only NSOs. Between October 2015 and December 2017, U-47700 was reported in nine deaths from Cook County, Illinois [87]. Due to these reports and its linkage to mortality, the US DEA announced emergency action to place U-47700 under Schedule I control on 11 November 2016 [88]. In addition to these actions in the US, international scheduling and bans on its manufacturing and exports were also implemented, ultimately leading to a substantial drop in the prevalence of U-47700. As a response to temporary drug scheduling, clandestine laboratories sought legal alternatives to U-47700, including other compounds within the benzamide and acetamide series to appear, which was also an added appeal to users of U-47700. However, none of the other substance of this subclass have reached the same level of popularity as U-47700 since its peak positivity. U-49900 was identified through online drug markets and research chemical vendor websites two months after the temporary scheduling of U-47700 by the DEA [45,89]. Based on what we have observed, it appears that the recreational drug markets were not willing to adopt another substance from the U-compounds as a viable and popular alternative to U-47700; at the time of peak popularity of U-47700, fentanyl analogs were not widely controlled under core-structure scheduling, so the majority of the market focus shifted back to fentanyl-based NSOs.

As discussed already, U-47700 was the most significant compound of this NSO subclass in terms of impact on drug markets, prevalence of use, and harm to end users [20]. The subjective effects of U-47700 have been described as desirable, with apparently short-lived euphoric and analgesic effects, in turn creating a desire to re-dose [20]. As noted previously, U-47700 is estimated to have an elimination half-life of 6 h in humans [48]. Besides the desired opioid-mediated effects, U-47700 can produce stupor, loss of consciousness, and respiratory depression, leading to death. There are reports of naloxone being used to successfully reverse U-47700 toxicity, as illustrated by the case of a 22 year old male who was found unconscious and apneic after using U-47700 that was purchased over the internet [90]. Table 4 provides an up-to-date summary of published case reports involving intoxications and fatalities where U-47700 was analytically-confirmed in toxicology specimens. In addition, there is one published case report involving U-49900. While U-49900 succeeded U-47700, it did not replace the market share that U-47700 developed, possibly due to reports that U-49900 failed to induce the desired euphoria and analgesia, even at high doses [89].

While there are some cases of intentional use of U-compounds, it is likely that most human exposures are unintentional or inadvertent. At least two case reports described descendants who specifically purchased “Norco” (i.e., hydrocodone/acetaminophen) pills on the street drug market, and after an apparent opioid overdose, only U-47700 was toxicologically confirmed [102]. Two individuals presented to the emergency department: one cyanotic and with agonal respirations and the other with anxiety, tremors, and drowsiness. They believe they had insufflated “synthetic cocaine” along with alcohol and alprazolam, but only U-47700 was detected in their urine [106]. U-47700 was included in the panel of opioid drugs detected for a study of donors enrolled in a clinically monitored pain management program between June 2016 and September 2017. The drug was detected in 1.39% of heroin-positive patients (*n* = 1152) but was not detected in the heroin-negative group; the patients exposed to U-47700 believed they had purchased heroin [107]. Another study identified the presence of U-47700 in urine samples collected from a cohort of patients presenting to an emergency department after reported heroin overdose; the findings support the conclusion that most recreational users are unaware of clandestine opioids that are being sold as a substitute for or diluent with heroin [77].

One of the many challenges associated with NPS use is the difficulty in understanding the prevalence of a particular compound when that substance typically is not included in the scope of routine toxicology testing. Toxicology testing methods can be modified or developed to detect NSOs as the compounds emerge on drug markets, but the actual extent of such testing is difficult to ascertain. There are a few reports that provide some insight into the popularity of these substances. In a quantitative 19 analyte panel of fentanyl analogs and novel synthetic opioids, U-47700 was the fourth most reported active substance, after 2-furanylfentanyl, carfentanil, and *para*-fluoroisobutyrylfentanyl/*para*-fluorobutyrylfentanyl. U-47700 was reported in 543 postmortem blood samples between October 2016 and September 2017, with a mean and median concentration of 143 ng/mL and 14 ng/mL, respectively (range: 0.20–3800 ng/mL). After the development of a seven-analyte panel that included U-47700, U-49900 and U-50488, 15 authentic postmortem blood samples were reported positive for U-47700, with a mean concentration of 214 ng/mL (range: 3.2–1448 ng/mL) [71].

The concentration range for U-47700 that is reported from the method development publications is consistent with the range of concentrations found in case reports involving U-compounds. U-47700 serum concentrations in non-fatal intoxications ranged from 7.6 to 394 ng/mL and the blood concentrations were reported at 94 and 282 ng/mL [51,52,99,101,102,104]. The blood concentrations for the U-47700 fatalities ranged from 7.8 to 3040 ng/mL, though most cases involved poly-drug use [25,26,46,48,78,80,91,92,93,94,95,96,97]. A few cases have reported U-47700 as the sole substance involved in both non-fatal intoxications and fatal drug overdoses. Even in these instances, the concentrations of U-47700 overlap for non-fatal and fatal cases. In non-fatal intoxications involving only U-47700, blood or serum concentrations ranged from 18 to 294 ng/mL [101,103,104]. In a case series reported by Mohr et al., the lowest concentration of U-47700, implicated as the sole source of intoxication, was 242 ng/mL [25]. In a case series of 26 deaths associated with U-47700 exposure reported from Germany, a single fatality involved only U-47700, with a concentration of 400 ng/mL, in addition to a 0.067% blood alcohol concentration (BAC). The median femoral blood concentrations for the German case series was 610 ng/mL (range: 27–2200 ng/mL) [24]. Case reports involving other U-compounds are sparse when compared to those involving U-47700. Because the concentrations of U-47700 in non-fatal and fatal cases overlap significantly, it is impossible to specifically identify a lethal concentration of the drug. U-49900 was reported in one fatality from December 2016, but this case also included the NSOs tetrahydrofuranylfentanyl (THFF) and methoxyphencyclidine (MeO-PCP) [105]. In postmortem toxicology, the significance of a reported drug concentration depends on many factors, including postmortem interval, other drugs detected and their concentrations, any degree of acquired opioid tolerance, co-morbid disease states, pharmacogenetic factors, and more. Such complexity makes it difficult to ascertain the specific toxicological significance of any particular drug at a specifically reported concentration. When all other significant pathological or toxicological findings can be ruled out, fatal toxicity due to exposure to a NSO, including U-compounds, may be the most likely cause.

## *7.* Conclusions

The data reviewed herein demonstrate that U-47700 is a potent MOR agonist, which poses profound risks to public health and safety. U-47700 and its analogs are classic examples of chemical entities that were diverted from the patent and medicinal chemistry literature for malicious purposes. Although a number of U-compounds have reached recreational markets worldwide, U-47700 is the only one of this group to achieve a level of prevalence sufficient for risk assessment by the World Health Organization. On the other hand, a number of structurally distinct non-fentanyl derived NSOs, including isotonitazene and brorphine, have recently gained traction in recreational markets [108,109]. The evolution of NSOs from substituted benzamides and acetamides to other structural templates reveals the continued diversification of compounds with opioid activity [110]. A number of lessons can be learned from the U-47700 experience. First, with regard to the pharmacological characterization of NSOs, newly developed in vitro methods can be utilized to rapidly characterize the drug potency and efficacy at the MOR, DOR, and KOR [44,111]. However, it is paramount to include in vivo animal studies for initial investigations because in vitro receptor activity can differ across various assay platforms and may not predict potency to induce analgesia or adverse effects, such as respiratory depression [44]. Secondly, newly synthetized drugs (and especially NSOs) often have a core structural template that allows for slight chemical modifications using simple synthetic schemes to produce compounds with a similar opioid activity and potency. This process has led to diversification of this subclass and the emergence of new U-compound analogs. Thirdly, with regard to the analytical detection of NSOs, newly developed methods for non-targeted drug detection and paired data-mining workflows should be sought or expanded. This would then allow for more readily available use in forensic laboratories involved with testing seized drug products and human toxicology samples [8]. The rapid synthesis and availability of purified reference materials remains a key factor for confirmatory testing, both for parent drugs and their metabolites. Finally, it is crucial that information about newly emerging narcotic drugs be shared in real-time with various stakeholders—including clinicians, chemists, toxicologists, and epidemiologists—to insure rapid and effective responsiveness to the continuing threat of NSOs. 

## Figures and Tables

**Figure 1 brainsci-10-00895-f001:**
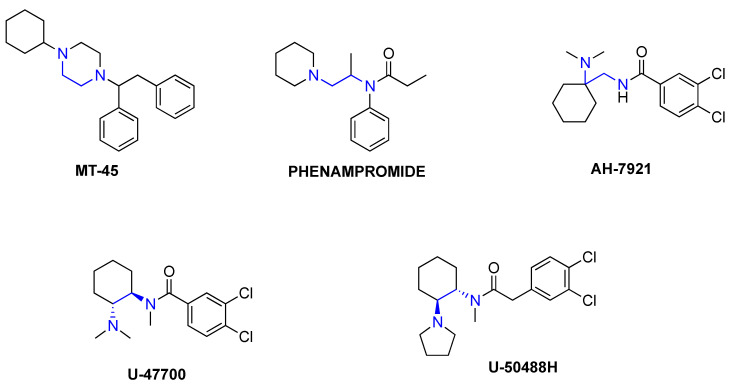
Chemical structures of novel synthetic opioids (NSOs) containing 1,2-ethylene diamine core components.

**Figure 2 brainsci-10-00895-f002:**
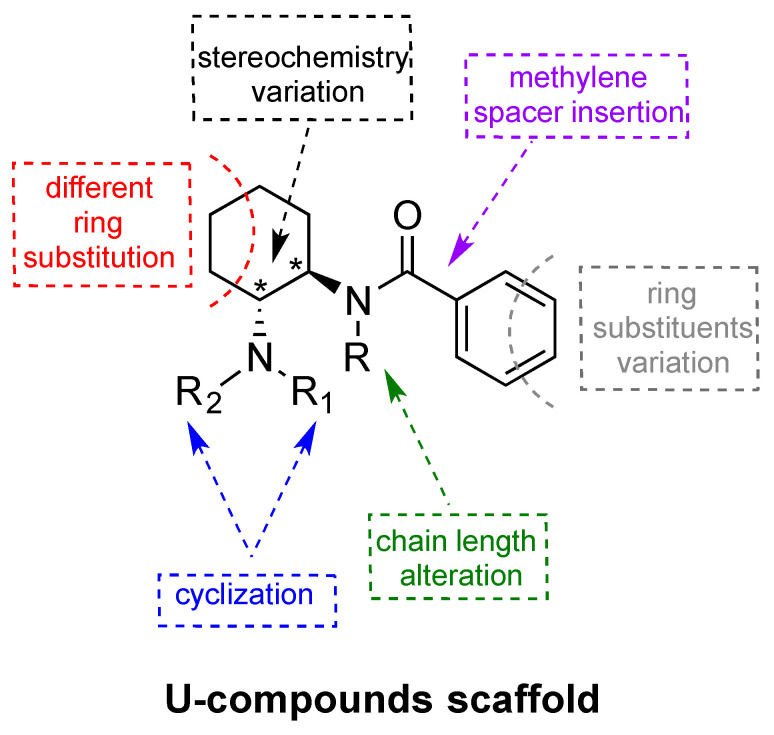
Generic U-compound scaffold and its potential for structural modification(s).

**Figure 3 brainsci-10-00895-f003:**
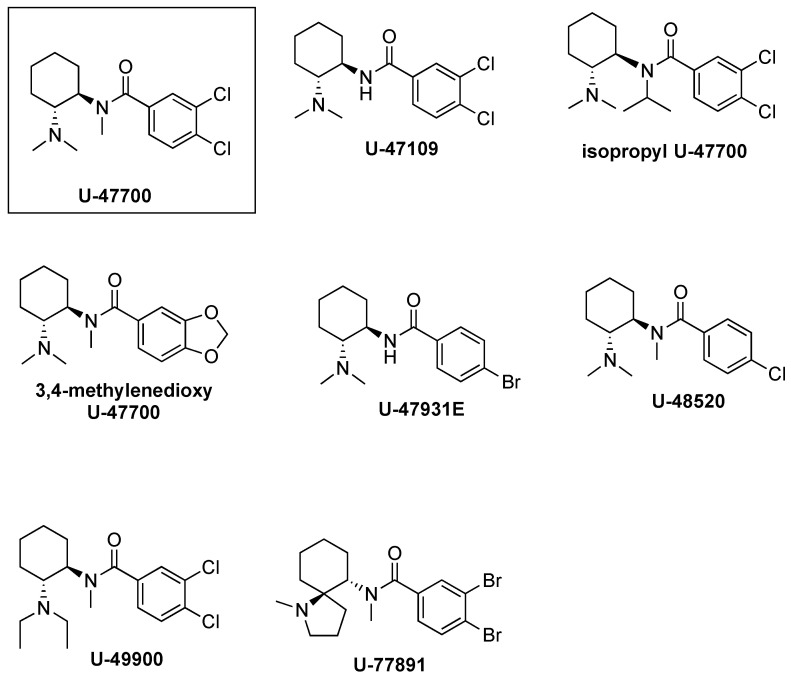
U-compounds without the methylene spacer, referred to as the U-47700 series.

**Figure 4 brainsci-10-00895-f004:**
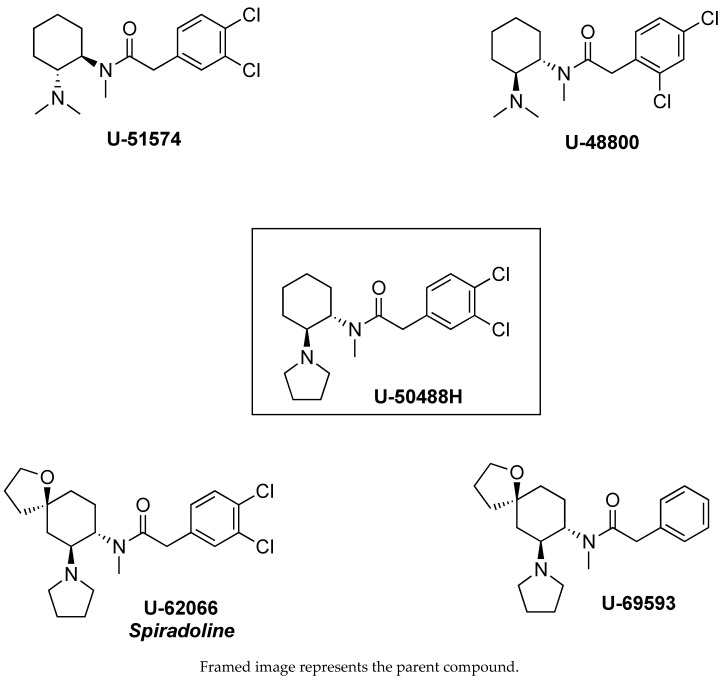
U-compounds containing the methylene spacer, referred to as the U-50488 series.

**Figure 5 brainsci-10-00895-f005:**
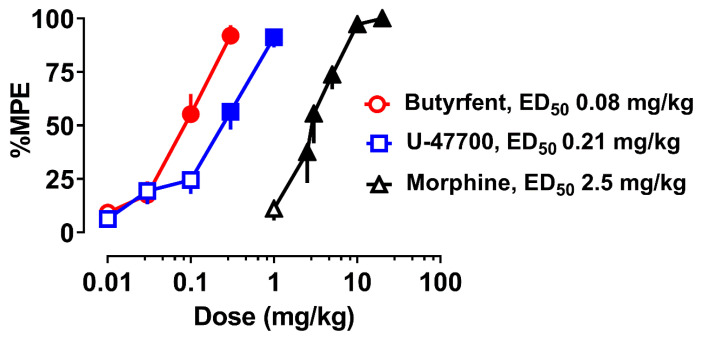
Effects of subcutaneous butyrylfentanyl, U-47700, and morphine on tail flick latency in the radiant heat tail flick test in CD-1 mice. Data are the mean ± SEM, expressed as percent maximum possible effect (%MPE), for *n* = 6 mice per group (from Baumann et al. [44]).

**Figure 6 brainsci-10-00895-f006:**
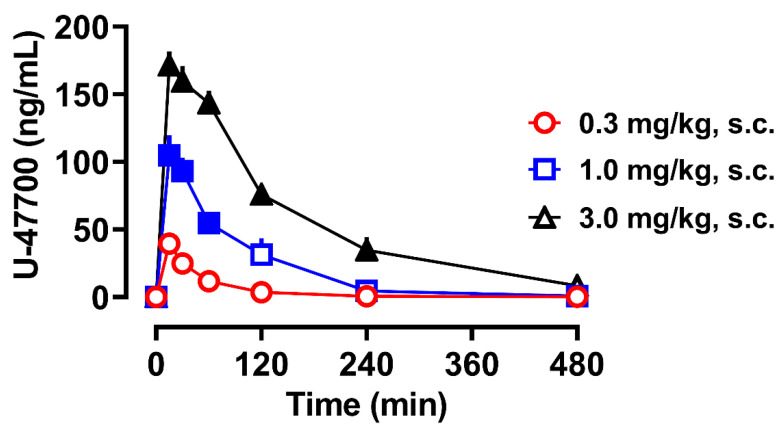
Plasma pharmacokinetics of U-47700 in male rats receiving subcutaneous injections of the drug. Data are the mean ± SEM expressed as ng/mL of plasma, for *n* = 6 rats per group (from Truver et al. [49]).

**Table 1 brainsci-10-00895-t001:** U-47700 series opioid receptor binding affinities (data from Loew et al. [35]).

^a^ Compound	MOR K_D_ (nm)	KOR K_D_ (nm)	Ratio MOR/KOR
U-47700 (3,4-dichloro-*N*-[(1*R*,2*R*)-2- (dimethylamino)cyclohexyl]- *N*-methylbenzamide)	5.3	910	0.006
U-47109 (3,4-dichloro-*N*-[(1*R*,2*R*)-2- (dimethylamino)cyclohexyl] benzamide)	59	910	0.065
U-48520 (4-chloro-*N*-[(1*R*,2*R*)-2- -(dimethylamino)cyclohexyl]- *N*-methylbenzamide)	200	2900	0.069
U-77891 3,4-dibromo-*N*-methyl-*N*-((5*S*,6*R*)- 1-methyl-1-azaspiro[4.5]decan-6-yl) benzamide	2	2300	0.001

^a^ Loew et al. [35] did not specify stereochemistry of compounds tested; here we employ IUPAC naming of the compounds.

**Table 2 brainsci-10-00895-t002:** U-50488 series opioid receptor binding affinities (data from Loew et al. [35]).

^a^ Compound	MOR K_D_ (nm)	KOR K_D_ (nm)	Ratio MOR/KOR
U-50488H (2-(3,4-dichlorophenyl)-*N*-methyl- *N*-[(1*S*,2*S*)-2-(pyrrolidin-1-yl) cyclohexyl]acetamide)	430	2.2	195
U-51574 (2-(3,4-dichlorophenyl)*-N*-[(1*R*,2*R*)-2- (dimethylamino)cyclohexyl]-*N*- methylacetamide)	220	71	3.1
U-62066 (2-(3,4-Dichlorophenyl)-N-methyl-N- [(5R,7S,8S)-7-pyrrolidin-1-yl-1- oxaspiro[4.5]decan-8-yl] acetamide)	210	2.5	85
U-69593 (2-phenyl-N-methyl-N-[(5R,7S,8S) -7-pyrrolidin-1-yl-1-oxaspiro[4.5] decan-8-yl] acetamide)	1700	7.2	236

^a^ Loew et al. [35] did not specify stereochemistry of compounds tested; here we employ IUPAC naming of compounds.

**Table 3 brainsci-10-00895-t003:** List of novel U-compounds identified in seized drug materials or toxicology samples as reported by the New Psychoactive Substances (NPSs) Discovery program), in chronological order.

Analyte	Sample Type	Date of Report
U-48800	Seized Material	March 2018
Isopropyl-U-47700	Biological Fluid	May 2018
Methylenedioxy-U-47700	Biological Fluid	May 2018
U-47931E	Seized Material	October 2018
Furanyl UF-17 *	Seized Material	June 2019
UF-17 *	Seized Material	June 2019
N-Methyl U-47931E	Seized Material	November 2019
3,4-Difluoro-U-47700	Seized Material	March 2020
N-Ethyl-U-47700	Seized Material	March 2020

* Contains a U-compound-like core structure, although not pharmacologically active.

**Table 4 brainsci-10-00895-t004:** Summary of reported toxicology cases involving U-47700 and U-49900.

Series/Case	U-Compound	Case Type	Brief History	Clinical Symptoms/Autopsy Findings	Drug Results (ng/mL unless Specified)	Reference
1.1	U-47700	PM	20 y/o M with hx of illicit drug use. Found with syringe still clutched in hand.	N/A	Peripheral Blood: U-47700 382, Amphetamine 12, caffeine	[25]
1.2	U-47700	PM	18 y/o M found unresponsive in bed. Had a hx of ADHD and illicit drug use. Paraphernalia was found on scene and presumptively identified to be butyrylfentanyl and U-47700.	N/A	Peripheral Blood: U-47700 17, EtOH 0.03 g/dL, butyrylfentanyl 26 Urine: carbamazepine, ibuprofen, iminostilbene, oxcarbazepine and suspected butyrylfentanyl Vitreous: EtOH 0.03 mg/dL	[25]
1.3	U-47700	PM	39 y/o M found unresponsive by wife. A syringe was found on the floor. Individual had a hx of ordering supplements and designer drugs off the internet.	N/A	Femoral Blood: U-47700 217, Mephedrone 22, etizolam	[25]
1.4	U-47700	PM	25 y/o M with hx of poly-substance abuse. Decedent was recently released from a halfway house. White powder found at the scene chemically confirmed to be U-47700.	Evidence of pulmonary edema—on scene.	Peripheral blood: U-47700 334	[25]
1.5	U-47700	PM	23 y/o M was found in the bathroom of a rehabilitation center with a ligature around his arm and a needle/syringe found on floor near decedent. A packet was found nearby containing a powdery substance labeled “U-47700”.	N/A	Peripheral blood: U-47700 252, Citalopram 43 Urine: cotinine, caffeine, nicotine, citalopram	[25]
1.6	U-47700	PM	29 y/o M was complaining of a headache and collapsed suddenly. EMS was called and resuscitation was unsuccessful.	Cerebral and pulmonary edema.	Blood: U-47700 453 Urine: cannabinoids, mitragynine	[25]
1.7	U-47700	PM	29 y/o M with a hx of drug abuse was released from a rehabilitation facility. Individual was found unresponsive with drug paraphernalia present: Rolled up 10-dollar bill and packets of white powders.	Evidence of pulmonary edema—on scene.	Peripheral blood: U-47700 242, THC-COOH 5.3	[25]
1.8	U-47700	PM	26 y/o M with hx of illicit substance use was found deceased at home. Paraphernalia was found on scene including: 5 syringes, 3 blue glass dropper bottles with liquid, 15 diphenhydramine tablets, and etizolam and Benadryl^®^ pills.	N/A	Blood: U-47700 103, Diphenhydramine 694 Urine: Diphenhydramine metabolites, 3-MeO-PCP Gastric Contents: Diphenhydramine and metabolites	[25]
1.9	U-47700	PM	21 y/o M with hx of drug abuse and convictions. Found with needle still penetrating injection site.	N/A	Aorta Blood: U-47700 299, tramadol < 250, alprazolam 47, lorazepam 11, 3-MeO-PCP 180 Urine: alprazolam, amphetamine, tramadol	[25]
1.10	U-47700	PM	32 y/o M was found deceased sitting in a chair.	N/A	Aorta Blood: U-47700 311, Oxycodone 11, venlafaxine 2600, O-desmethylvenlafaxine 380 Liver: Venlafaxine 7.5 mg/kg, O-Desmethylvenlafaxine 0.87 mg/kg	[25]
1.11	U-47700	PM	24 y/o M was reported to be using U-47700 that was obtained over the internet.	N/A	Aorta Blood: U-47700 487, etizolam 86, chlorpheniramine < 250, diphenhydramine 250	[25]
1.12	U-47700	PM	24 y/o M with a hx of substance abuse was found unresponsive.	N/A	Aorta Blood: U-47700 59, 4-ANPP, quinine, furanylfentanyl	[25]
1.13	U-47700	PM	36 y/o M was found unresponsive in a bathroom with a syringe cap in his mouth.	N/A	Aorta Blood: U-47700 135, furanylfentanyl 26, EtOH, 4-ANPP, and quinine	[25]
1.14	U-47700	PM	33 y/o M with hx of cocaine and heroin use.	N/A	Aorta Blood: U-47700 167, furanylfentanyl 56, morphine 48, 4-ANPP, 6-MAM, and quinine	[25]
1.15	U-47700	PM	29 y/o M found unresponsive of a suspected heroin overdose.	N/A	Aorta Blood: U-47700 490, furanylfentanyl 76, 4-ANPP and quinine	[25]
1.16	U-47700	PM	40 y/o M with hx of heroin/opioid abuse.	N/A	Aorta Blood: U-47700 105, furanylfentanyl 2.5, 4-ANPP and quinine	[25]
2.1	U-47700	PM	27 y/o M was found dead at home. It was believed that the individual snorted mirtazapine (prescribed) and had a hx of use of cannabis, ketamine, MCAT, and “legal highs”.	No natural disease or cause of death was found.	Femoral blood: U-47700 1460, quetiapine < 50 Amphetamine < 100 Naproxen < 0.8 mg/L Urine: quetiapine, amphetamine, amitriptyline, mexedrone, ketamine	[91]
3.1	U-47700	PM	30 y/o M found deceased in his home after inhaling fumes of a powder burned on aluminum foil. A recently delivered envelope from China containing 36 g of a white powder, a digital scale and a spoon were also found in the room. Individual did have a hx of drug abuse and experimenting with drugs purchased over the internet. Powder found on scene was determined to be fentanyl and U-47700.	No injection sites or traumatic injuries were found Pulmonary edema observed Only external body examination was ordered.	Subclavian blood: U-47700 13.8, fentanyl 10.9, sertraline 180 Urine: U-47700 71	[92]
4.1	U-47700	PM	46 y/o M had a hx of ingesting substances he purchased through the mail. Individual had ingested the substance previously and passed out on the floor but was able to be awaken with no need for medical assistance. Individual was then found unresponsive the next morning in his bedroom. EMS was called and the individual was declared deceased on scene. An envelope labeled “U-47700” and a straw were found in the bedroom. The compound was obtained through the mail 3 days prior to incident.	No needle puncture sites identified Enlarged heart (570 g) with Dilation of the right ventricle Lungs were mildly edematous and congested (right 500 g, left 640 g)–microscopically lungs showed edema, congestion and expanded alveoli with terminal clubbing of septa Liver and spleen were enlarged (2310 and 300 g, respectively).	Peripheral Blood: U-47700 190, alprazolam 120, nordiazepam < 50, doxylamine 300, diphenhydramine 140, ibuprofen 2.4 mg/L, salicylic acid < 20 mg/L and THC-COOH 2.4 Central Blood: U-47700 340 Liver: U-47700 1700 ng/g Vitreous: U-47700 170 Urine: U-47700 360 Gastric Contents: U-47700 < 1 mg	[80]
5.1	U-47700	PM	Individual found dead in bed.	Cerebral/lung edema.	Femoral Blood: U-47700 525, diphenidine ~1.7, methoxphenidine ~26, ibuprofen 1.8 mcg/mL, naloxone 1.9 Heart Blood: U-47700 1347 Urine: U-47700 1393 Kidney: U-47700 2.7 ng/mg Liver: U-47700 4.3 ng/mg Lung: U-47700 3.2 ng/mg Brain: U-47700 0.97 ng/mg	[93]
5.2	U-47700	PM	Individual found dead in bed.	Cerebral/lung edema.	Femoral Blood: U-47700 819, diphenhydramine 45, methylphenidate 2.5 Heart Blood: U-47700 1043 Urine: U-47700 1848 Kidney: U-47700 1.4 ng/mg Liver: U-47700 3.1 ng/mg Lung: U-47700 2.4 ng/mg Brain: U-47700 1.1 ng/mg	[93]
6.1	U-47700	PM	A 28 y/o M was found deceased at home with dried froth on his mouth and nose. Several plastic baggies labeled as “U-47700”, “etizolam”, “butyrylfentanyl” and “4-2CO-MET” were found in the room with other drug paraphernalia.	Pulmonary edema, hypertensive-type cardiomegaly, and chronic active hepatitis with steatosis.	Femoral Blood: U-47700 189, oxycodone 67, dextro/levomethorphan 17	[94]
6.2	U-47700	PM	An 18 y/o M was found deceased on his bedroom floor. He had bloody fluid emanating from his nose. Baggies labeled as “U-47700” and “etizolam” were found at the scene with other drug paraphernalia.	Pulmonary edema, cerebral edema, asthma, and focal subgaleal hemorrhage consistent with a terminal fall.	Femoral Blood: U-47700 547, etizolam	[94]
7.1	U-47700	PM	34 y/o M with history of Illicit drug use, as well as depression and bipolar disorder, found unresponsive in a bathroom. Drug paraphernalia, including needles and an empty plastic bag labeled “3-FPM”, were located near the decedent.	Pulmonary edema and congestion.	Femoral blood: U-47700 360, 3-fluorophenmetrazine 2400, amitriptyline 440, nortriptyline 290 methamphetamine < 40, amphetamine 70, diazepam 200, nordiazepam 180, temazepam 11, flubromazolam, delorazepam	[95]
8.1	U-47700	PM	51 y/o M was found unresponsive in a hotel bathroom. Paraphernalia was found on scene: a spoon, white powder and a syringe.	Brain swelling (1570 g), lung hyperhydration (right lung 905 g and left lung 665 g) and hyperemia of the inner organs (heart 375 g, spleen 185 g, liver 1635 g, right kidney 145 g, left kidney 210 g). Injections site were observed on the right and white crumbs in the gastric contents.	Femoral Blood: U-47700 290, oxycodone 50, noroxycodone 40 Heart Blood: U-47700 1290 Liver: U-47700 9.9 ug/g Urine: U-47700 240 Gastric Contents: U-47700 570 Bile Fluid: U-47700 2300 Cerebrospinal fluid: U-47700 400 Hair: U-47700 0.14 ng/mg	[96]
9.1	U-47700	PM	A 26 y/o M with history of illicit drug use found deceased in his bedroom. A foam cone was noted. A box containing drug paraphernalia was located at the scene.	Moderate congested and edematous lungs and urine retention.	Femoral blood: U-47700 400 Heart blood: U-47700 260 Vitreous Fluid: 90 Brain: U-47700 0.38 ng/mg Liver: U-47700 0.28 Urine: U-47700 4600	[26]
10.1	U-47700	PM	A 28 y/o M with history of illicit drug use was found deceased at home. Decedent was known to use methamphetamine and had told a friend at times he took a “benzo” believed to be etizolam.	N/A	Peripheral blood: U-47700 330, Diclazepam 70, flubromazepam 10, methamphetamine 290, amphetamine 150, DOC	[97]
11.1	U-47700	PM	A 63 y/o M with a history of drug abuse was found deceased at home by girlfriend.	Autopsy findings included cirrhosis, arterionephrosclerosis, intact artificial aortic and mitral values, pleural and pericardial adhesions, and cardiomegaly with left ventricular hypertrophy.	Peripheral blood: U-47700 24, Cyclopropylfentanyl 31.5, EtOH 0.025 g/dL, cocaine 25, BZE 58	[98]
11.2	U-47700	PM	A 57 y/o M with a history of drug abuse became unresponsive in a vehicle. The decedent was transferred to a hospital via EMS but pronounced dead on arrival.	Autopsy findings included pulmonary edema.	Postmortem blood: U-47700 7.8, Cyclopropylfentanyl 18.5 cocaine 130, BZE 910, fentanyl 6.2, norfentanyl 5.5	[98]
12.1	U-47700	PM	The M decedent, who had a history of drug addiction, was found deceased in his apartment. Two amber glass bottles of nasal spray containing a transparent liquid and a plastic bag containing a white powder were located at the scene, in addition naloxone.	General pulmonary edema noted at autopsy.	Blood: U-47700 380 Urine: U-47700 10400	[78]
13.1	U-47700	PM	A 24 y/o M suffered apnea after the consumption of U-47700 and flubromazepam. After hospital admission, hypoxic cerebral damage and severe brain edema were stated. After six days in the hospital, mechanical ventilation was discontinued.	N/A	Admission Serum: U-47700 370, Flubromazepam 830, hydroxyflubromazepam 160, pregabalin 1.7 mcg/mL Serum + 9 hrs: U-47700 37, Flubromazepam 370, hydroxyflubromazepam 87 pregabalin 1.3 mcg/mL Serum + 24 hrs: U-47700 6.3, Flubromazepam 530, hydroxyflubromazepam 120 pregabalin 0.29 mcg/mL	[48]
14.1	U-47700	PM	A 19 y/o M was found deceased at home. Paroxetine tablets were located at the scene.	Autopsy findings were not specific.	Peripheral Blood: U-47700 3040, acetaminophen 700 oxazepam 30, dextromethorphan 20, paroxetine 170, alpha-PHP	[46]
15.1	U-47700	Intoxication	29 y/o M found unresponsive after intravenous injection of U-47700. Regained consciousness before being transferred to the ED. Individual admitted to purchasing U-47700 and phenazepam over the dark net, and the phenazepam was ingested a few days prior to the ED visit.	At ED: no complaints other than feeling thirst BP 157/105 mmHg HR 112 bpm Creatinine 1.4 mg/dL Was discharged after 3 h.	Serum: U-47700 240, phenazepam 1400 Urine: U-47700 Positive, phenazepam	[99]
16.1	U-47700	Intoxication	28 y/o M found unconscious in the driver seat of a vehicle. A needle was observed in the individual’s lap and a further search of the car yielded two clear plastic baggies. Individual was then transported to the ED and eventually released. A blood sample was provided over 2 h after the incident.	Field sobriety tests and DRE evaluations were not performed due to individual being unconscious.	Antemortem Blood: U-47700 (>100 pg/mL), alprazolam 55, fentanyl < 0.5, carfentanil, furanylfentanyl, para-fluoroisobutyrylfentanyl, N-desmethyl U-47700	[100]
17.1	U-47700	Intoxication	A 23 y/o F presented to the ED after insufflating and injecting “U4”. She was cyanotic with respiratory depression and responded to naloxone in the field.	N/A	Serum: U-47700 394 Urine: U-47700 228	[101]
18.1	U-47700	Intoxication	A 41 yr old woman who presented to the ED with pinpoint pupils, respiratory depression, and depressed consciousness after consuming three pills of street purchased “Norco”.	N/A	Serum: U-47700 7.6, fentanyl 15.2, acetaminophen 10 mg/L, BZE 46, gabapentin 350, hydrocodone 107, sertraline 15	[102]
19.1	U-47700	Intoxication	A M in his 30s was found unresponsive in his residence and was transported to the ED in a comatose state with severe respiratory depression. U-47700 was chemically identified in a white powder found at the scene.	N/A	Admission blood: U-47700 94 Blood + 24 hrs: U-47700 3.0 Admission Urine: U-47700 5.2 Urine + 24 hrs: U-47700 5.4 Pubic hair: U-47700 3.02 ng/mg	[51]
20.1	U-47700	Intoxication	A 28 y/o M was found unresponsive by roommate after ingesting “Norco” he had purchased from a drug dealer. When EMS arrived, he was cyanotic with respiratory depression and did not respond to naloxone.	At ED, patient had systolic blood pressure of 95 mmHg, diastolic blood pressure of 70, oxygen saturation of 84%, and respiration rate of 10 breaths/min. Physical examination noted pinpoint pupils.	Serum: U-47700 18, acetaminophen < 10 mcg/mL Urine: U-47700 130	[103]
21.1	U-47700	Intoxication	A 26 y/o M found unconscious at home. Individual was found with 3 bags containing white powder, with one labeled as MMB CHMINACA.	On scene: Depressed respiration noted, Score of 3 on the GCS ED: hemodynamically stable, bp 90/50 mmHg, pulse rate 50 bp/m, unconscious and required artificial ventilation for 12 h. Patient was discharged 40 h after admission.	Serum: U-47700 351, U-47700-N,N-didesmethyl, U-47700-N-demethyl-hydroxy, U-47700-N,N-didesmethyl-hydroxy, THC 3.3, THC-COOH 121.6, clonazolam 6.8, citalopram < 5, midazolam < 10 Urine: U-47700, citalopram, midazolam, clonazolam, THC	[52]
22.1	U-47700	Intoxication	A 17 y/o M became apneic and unresponsive en route to school after insufflating a white substance he believed to be “crushed Xanax”. Naloxone was administered and the individual was transported to an ED with signs consistent with opioid toxidrome.	N/A	Blood: U-47700 282	[104]
23.1	U-49900	PM	A 31 y/o M was found vomiting and convulsing in his residence by a family member. The decedent has a history of schizophrenia, bipolar disorder, and illicit drug use. Emergency rescue personnel were called but resuscitation efforts were unsuccessful, despite use of naloxone. Drug paraphernalia was located on scene, including a needle and two baggies containing a suspected controlled substance; one baggie was marked as “THFF”. Chemical testing on the powder identified the presence a smaller amount of U-49900 in addition to the tetrahydrofuranylfentanyl.	Pulmonary congestion and edema, in addition to cerebral edema, noted at autopsy.	Postmortem blood: U-49900 1.5, Tetrahydrofuranylfentanyl 339, methoxyphencyclidine 1.0	[105]

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
