# Peer review of "U-47700 and Its Analogs: Non-Fentanyl Synthetic Opioids Impacting the Recreational Drug Market"

_brainsci, 2020, doi:10.3390/brainsci10110895_

Round 1

Reviewer 1 Report

In this review, authors described all the crucial aspects concerning the prototype of non-fentanyl novel synthetic opioid (NSO) compounds, i.e., U-47700, and its analogs, thus providing a comprehensive overview of these new drugs that are causing serious health concern worldwide.

The topic is hot and the manuscript is very well structured and fluent to read. After a focused introduction presenting the “U-compounds”, their chemistry and pharmacology are discussed with the support of a series of clear figures and useful tables, followed by a description of past and current popularity and availability on the drug market. Highly appreciable are the exhaustive description of the analytical identification methods of NSO and the presentation of reported toxicology cases involving U-47700 (Table 4). Actually, the accurate series of figures displaying NSO chemical structures and potential for chemical modifications as well as table showing opioid receptor binding affinities are all very useful for readers, including those not strictly familiar with novel synthetic psychoactive substances.

I have found reading this review very pleasant, and have no major issue to highlight. The only suggestion I wish to give the authors is to include in their paper the following two recent reviews dealing with U-47700 and analogs:

Rambaran KA, Fleming SW, An J, Burkhart S, Furmaga J, Kleinschmidt KC, Spiekerman AM, Alzghari SK. U-47700: A Clinical Review of the Literature. J Emerg Med. 2017 Oct;53(4):509-519. doi: 10.1016/j.jemermed.2017.05.034..

Beardsley PM, Zhang Y. Synthetic Opioids. Handb Exp Pharmacol. 2018;252:353-381. doi: 10.1007/164_2018_149.

and to clarify which innovative aspects the present manuscript presents with respect to the two above and the other reviews published over the last 3 years (Armenian et al. 2018; Sharma et al. 2019; Kyei-Baffou and Lindsley 2020). This would help the readers to greatly appreciate the original aspects of the present review.

Author Response

We thank the reviewer for the favorable assessment of our paper.  In response to his minor concerns we have edited the MS as follows:

1] We added the suggested references of Beardsley and Zhang, 2018 [now ref.#6] and Rambaran et al., 2017 [now ref#28].

2] We have now added text in the introduction to state that:  "Published reviews about U-47700 have covered its history and diversion [20], synthetic schemes and structure activity relationships (SAR) [21], as well as clinical symptoms [28]. Here, we provide an updated review of the medicinal chemistry, preclinical pharmacology, clandestine availability, methods for detection, and forensic toxicology of U-47700 and selected analogs. A complete summary of the human casework involving U-47700 intoxication and death is included."

Reviewer 2 Report

Dear Authors,

I hope that you are all in good health during these difficult days.

The paper "U-47700 and its analogs: non-fentanyl synthetic opioids impacting the recreational drug market" aims to compile the information available in the literature on the pharmacology and toxicology of the compound U-47700 and its analogs. The work is well written and organized. The references (those I have checked) appear to be valid. The information provided by this work, although limited, I believe is beneficial to the body of knowledge available about the effect of this particular compounds.

I am not sure that I agree with all of the authors' statements, although I believe that this is part of the scientific discussion, especially on a subject where there is not so much evidence.

I would suggest to the authors that in Table 4 they could eliminate the reference to U-49900 as all lines, except the last one, refer to U-47700. Similarly, in order to make the information more accessible, I think it is convenient to eliminate some rows, for example, in Table 1 for the compound U-49900 no affinity data is reported, therefore, I think it would be convenient to briefly indicate this in the table caption and eliminate that row from the table. The same could be valid for other compounds in that table and for some of those in Table 2.

Author Response

We thank the reviewer for the positive assessment of our paper.  To address the minor concerns of the reviewer we have revised the MS as follows.

1] We removed rows/compounds from Table 1, where no receptor binding data were available.

2] We removed rows/compounds from Table 2, where no receptor binding data were available.